# Exploring Biodiversity and Food Webs in Sulfur Cave in the Vromoner Canyon on the Greek–Albanian Border

Serban M. Sarbu [1,2,3], Traian Brad [3,*], Raluca I. Băncilă [1,3,4,*] and Andrei Ştefan [3,5]

1 "Emil Racoviţă" Institute of Speleology, Calea 13 Septembrie, Nr. 13, 050711 Bucharest, Romania; serban.sarbu@yahoo.com
2 Department of Biological Sciences, California State University, Chico, CA 95929, USA
3 "Emil Racoviţă" Institute of Speleology, Str. Clinicilor, Nr. 5-7, 400006 Cluj-Napoca, Romania; andrei.stefan@antipa.ro
4 Faculty of Natural and Agricultural Sciences, Ovidius University Constanța, Aleea Universităţii, Nr. 1, 900470 Constanța, Romania
5 "Grigore Antipa" National Museum of Natural History, Şos. Kiseleff, Nr. 1, 011341 Bucharest, Romania
* Correspondence: traian.brad@acad-cj.ro (T.B.); bancila_ralucaioana@yahoo.com (R.I.B.)

**Abstract:** Sulfidic caves support diverse and abundant subterranean communities, including numerous endemic species and complex food webs, though the full extent of species diversity and resource utilization in these ecosystems remains largely unexplored. This paper presents the results of biological surveys conducted from 2023 to 2024 in Sulfur Cave, located in the Vromoner Canyon on the Greek–Albanian border, focusing on microbial, vertebrate, and invertebrate communities and investigating the structure of the subterranean food web. The microbial communities from the different biofilms are dominated by chemosynthetic sulfur-oxidizing microorganisms, specifically filamentous bacteria such as *Thiotrix* and *Beggiatoa*. Two species of fish, an eel (*Anguilla* sp.) and a Cyprinid (*Alburnoides* sp.), and six bat species from three families (Rhinolophidae, Miniopteridae, and Vespertilionidae) were documented. The invertebrate fauna includes five aquatic species, 25 terrestrial species, and four amphibiotic species. Among these, eight species are endemic, and seven species exhibit troglomorphic traits. Stable isotope analysis showed light carbon and nitrogen values for the terrestrial and aquatic invertebrates, suggesting that subterranean communities rely on food produced in situ by chemoautotrophic microorganisms. Our results identified cave areas of significant biological relevance and provided reference data to inform conservation actions aimed at preserving the biodiversity of this sulfidic cave.

**Keywords:** sulfidic subterranean ecosystems; thermo-mineral cave; chemoautotrophy-based food web; hotspot of subterranean diversity

## 1. Introduction

Life without sunlight was not considered possible prior to the discovery of ecosystems solely based on chemoautotrophy, such as deep-sea hydrothermal vents, in 1977 [1]. The subsequent discovery of a chemoautotrophic sulfidic groundwater ecosystem in Movile Cave, Romania [2], and later in other sulfidic caves, showed that large amounts of organic material produced in situ by autotrophic sulfur- and methane-oxidizing microorganisms can often support abundant and diverse subterranean invertebrate communities including numerous endemic species [3]. All the sulfidic underground ecosystems explored to date emerged as hotspots of subterranean biodiversity [4–6] and proved able to support complex subterranean food webs. Chemolithoautotrophs, often deemed less efficient than phototrophs, were traditionally not considered significant primary producers, but they play a crucial role in sustaining many ecosystem-level processes in the absence of light and photosynthesis in subsurface environments [5]. Although they are very rare, ecosystems based on chemoautotrophy are of particular interest to researchers as they may represent

analogs for Earth's earliest biological communities or for possible extraterrestrial life [7]. The enrichment of methane- and sulfur-oxidizing bacteria to produce microbial protein enables the production of alternative proteins with a reduced environmental impact compared to plant- or animal-based sources [8]. Despite the importance of chemolithoautotrophy, our understanding of cave chemosynthetic systems remains scarce.

Several sulfidic caves have recently been discovered and explored in Albania and northwestern Greece (Figure 1A) by teams of Italian [9] and Czech speleologists [10,11]. Thermo-mineral sulfidic water emerges from springs in the Vromoner and in the Pixaria canyons, which were carved by the Sarandaporo River in limestone outcrops near the border between Greece and Albania [12]. One of the largest springs is located in the deep recesses of Sulfur Cave, which straddles the border between the two countries in the Vromoner Canyon.

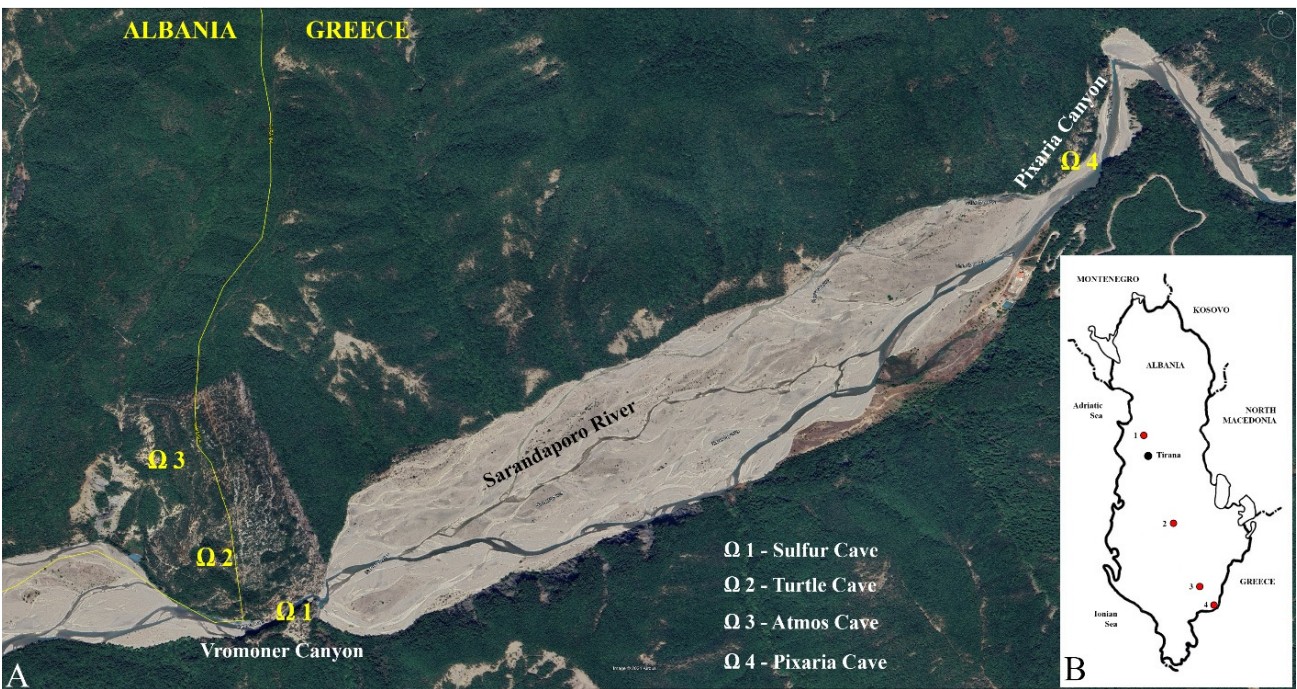

**Figure 1.** Location of sulfidic caves in Vromoner and Pixaria Canyons along the Sarandaporo River in NW Greece and SE Albania (**A**) and the location of the four sulfidic cave areas in Albania (**B**), i.e., (1) Zalles Cave, (2) Holtas Canyon, (3) Langarica Canyon, and (4) Sarandaporo River area on the border with Greece.

Subterranean sulfidic streams and large lakes have also been documented in Turtle Cave and Atmos Cave, located in the same limestone outcrop (Figure 1A). Due to the scarcity of studies, the full extent of species diversity or resource utilization of these chemoautotrophy-based ecosystems remains limited. In Sulfur Cave, for example, early biological observations were performed by members of the Czech Speleological Society in 2021, who reported an unusual abundance of fauna including a dense population of small flies and a large section of the cave walls covered by an extensive spider web with a multitude of spiders lurking in small funnel-shaped cavities within the web. The spiders collected during the early exploration expeditions were identified by Vlastimil Rýžiÿka. Notably, these caves are not covered by protected areas or legally protected, and some areas have already been affected by anthropogenic activity. Efforts are needed to engage local stakeholders to foster conservation actions for the preservation of these unique cave ecosystems and the invaluable biological communities they host.

The results of the observations and biological surveys conducted in Sulfur Cave between 2023 and 2024 are presented here. The microbial, vertebrate, and invertebrate communities were surveyed, and the structure of the subterranean food web was investi-

gated. One of the goals of this research is to identify areas of biological relevance and to provide important reference data to inform conservation actions aimed at preserving the biodiversity of this subterranean sulfidic ecosystem.

## 2. Materials and Methods

### 2.1. Sulfur Cave

Sulfur Cave is a 520 m-long hypogenic cave (Figure 2) located in the Vromoner Canyon, on the border between Greece and Albania (40.0961 N, 20.6789 E). It has a small entrance that opens into the Sarandaporo River, a tributary of the Aaos/Vjosa River. The main gallery increases in size, culminating in a large passage called Vesmír Dom (Universe Hall) and narrowing again in the most remote cave passage. The cave is traversed by a sulfidic subterranean stream that forms a lake near the entrance. The water emerges from springs located in the deepest recesses of the cave and flows all the way to the cave's entrance, where the water is discharged into the Sarandaporo River. The subterranean stream bifurcates in the Vesmír Dom and then rejoins to create a loop and flows through a small diverticulum of the main gallery, forming a chapel-like area known as Sulfur Chapel. In the Vesmír Dom, the largest room of the cave, the sulfidic stream reaches a width of 2–3 m, and the water is slightly milky due to the particles of elemental sulfur resulting from the oxidation of hydrogen sulfide ($H_2S$). Walls are often dry in this section of the cave because of the large deposits of hydrophilic anhydrite ($CaSO_4$) resulting from sulfuric acid speleogenesis (SAS). In the most remote part of the cave, there is a pool called Blue Eye, which represents one of the strongest sulfidic springs in Sulfur Cave.

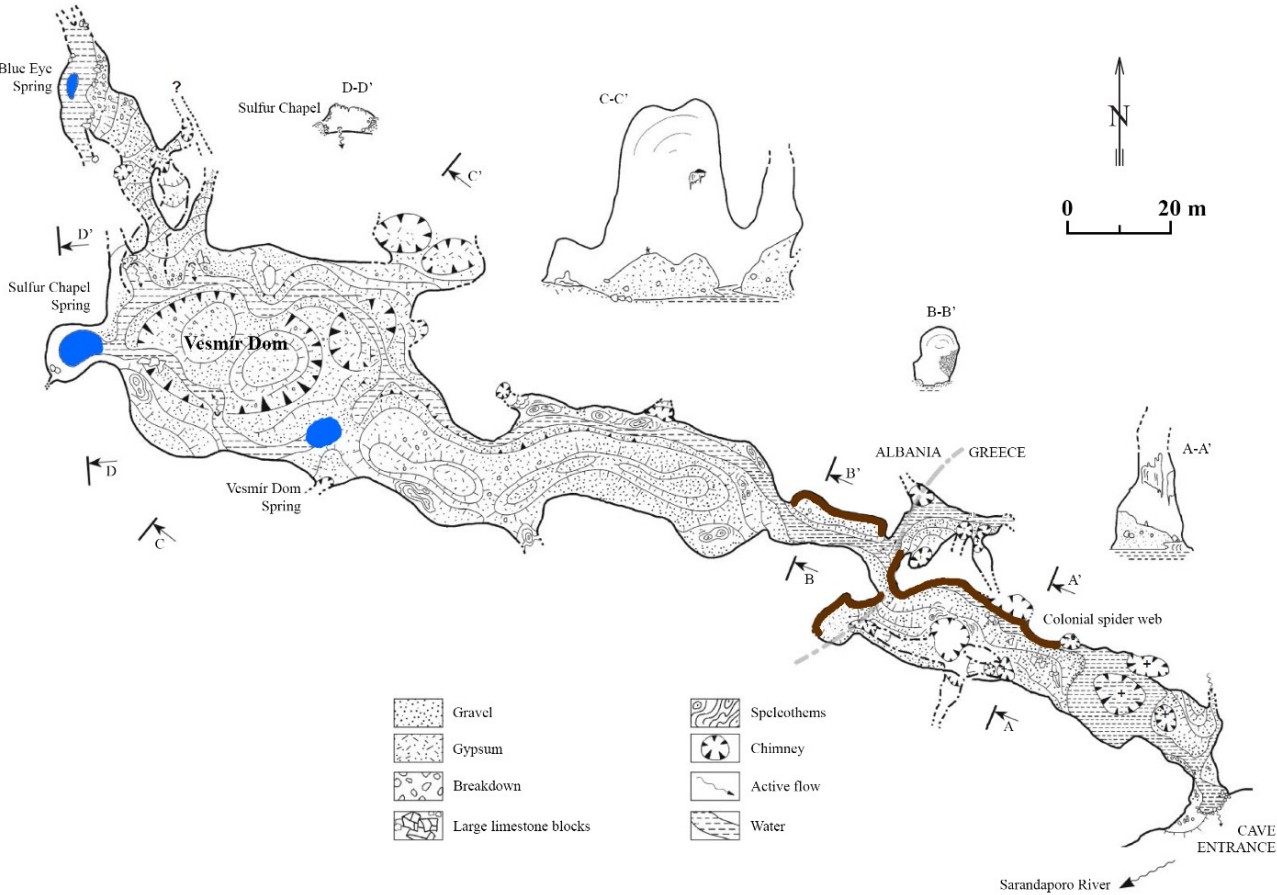

**Figure 2.** Plan of Sulfur Cave, with depiction of the main sulfidic springs (blue areas), and the large spider web in the vicinity of the cave entrance (brown areas) (modified from [10]).

In a few locations, small amounts of freshwater seep in from the epikarst, and the cave walls are moist and devoid of anhydrite deposits. The cave floor is primarily composed of

fluvial gravel and gypsum deposits, with large limestone boulders present in some areas. Accumulations of guano on the cave floor are rare. The concentration of $H_2S$ in the springs in Sulfur Cave can be as high as 65 mg/L; the water temperature is constant at 27 °C, with no significant annual variations (unpublished); and the pH of the water is neutral due to the strong buffering capacity of the carbonate bedrock. The cave atmosphere is highly enriched in $H_2S$, reaching levels up to 14 ppm in the vicinity of the strong emissions of sulfidic water within the cave, due to degassing that takes place at the surface of the springs. Air temperatures of up to 29 °C have been recorded in the upper cave passages. High water levels of the Sarandaporo River can flood the cave at a distance of up to 160 m from the cave entrance, as occurred in November 2023, depositing surface alluvial materials [11].

*2.2. Sampling*

2.2.1. Microbial Communities

To describe the microbial communities, biofilms were collected from above and below the water table. Thick white filamentous biofilms were harvested from sediments at the bottom of the sulfidic stream. In addition, samples of biofilms from the mud banks of small pools located in Sulfur Chapel, which are covered by thick brown biofilms, were also collected along with samples of biofilms from the nearby cave walls, which were covered by slightly blue biofilms. Sterile plastic tubes and pipettes were used for harvesting biofilms, and the samples were placed in sterile plastic tubes and kept frozen until analysis.

2.2.2. Vertebrate and Invertebrate Communities

For vertebrates, visual searches of bats and fish were performed for their presence or indirect traces (i.e., skulls), and the sounds produced by the bats were recorded. Invertebrates were sampled visually and manually using tweezers, pipettes, small paintbrushes dipped in ethanol, and occasionally a small plankton net. A UV light was used to search for scorpions. Samples were preserved in 70% ethanol for taxonomic identification and 96% ethanol for genetic and stable isotope analyses, and they were stored at −20 °C prior to processing. To minimize the negative impact on the cave populations, a limited number of specimens was collected for each invertebrate species encountered. All specimens were identified to the lowest possible taxonomic level.

The density of the larvae of *Chironomus* sp. (Diptera, Chironomidae) and *Contacyphon palustris* (C. G. Thomson, 1855) (Coleoptera, Scirtidae), found on the rocky sediments, was estimated by counting individuals within a 15 × 15 cm quadrant placed at ten randomly selected locations for each species (Figure 3A). Two high-resolution pictures of each quadrat were taken, and the number of individuals per quadrat was determined from the pictures.

2.2.3. Carbon and Nitrogen Stable Isotope Analysis

Organic samples were collected manually using fine tweezers and were dried at 60 °C for 24 h. The stable isotope ratios ($\delta^{13}C$ and $\delta^{15}N$) were determined at the Stable Isotope Lab, University of New Mexico, Albuquerque, NM, USA.

2.2.4. Molecular Identification

Morphological assignment of taxa was complemented by molecular identification. This was particularly useful for sub-adult invertebrates. Whole specimens were placed in 96% ethanol and kept at ambient temperature prior to processing in the lab. DNA extraction was carried out overnight using the QIAamp DNA Mini Kit (Qiagen, Hilden, Germany), and a fragment of the mitochondrial COI gene (subunit I of cytochrome c oxidase) was amplified using the universal primer pair LCO1490 (GGTCAACAAATCATAAA-GATATTGG)/HCO2198 (TAAACTTCAGGGTGACCAAAAAATCA) [13] or the denatured pair ACOIAF (CWAATCAYAAAGATATTGGAAC)/ACOIAR (AATATAWACTTCWGGGT-GACC) [14]. A PCR was performed in a 40 µL mixture containing 1X AccuStart II PCR ToughMix® (Quantabio, MA, USA), 1X GelTrack Loading Dye (Quantabio, MA, USA), 0.15 µM of each primer, and up to 20 ng/µL of DNA. The cycling conditions consisted of

5 cycles of denaturation at 94 °C for 30′, primer annealing at 45 °C for 30′, and extension at 72 °C for 50′ followed by 35 cycles of denaturation at 94 °C for 30′, primer annealing at 51 °C for 30′, and extension at 72 °C for 50′. Amplification was confirmed by electrophoresis on 1.5% (*w/v*) agarose gel, and PCR products were purified and Sanger-sequenced (Macrogen, Amsterdam, The Netherlands). The resulting chromatograms were visually inspected in Chromas v.2.6.6 (Technelysium Ltd., South Brisbane, Australia), low-quality ends trimmed, and assembled in CodonCode Aligner v.3.7.1 (CodonCode Corporation, MA, USA). The manually curated DNA sequences were checked against publicly available databases such as BOLD [15] and GenBank [16].

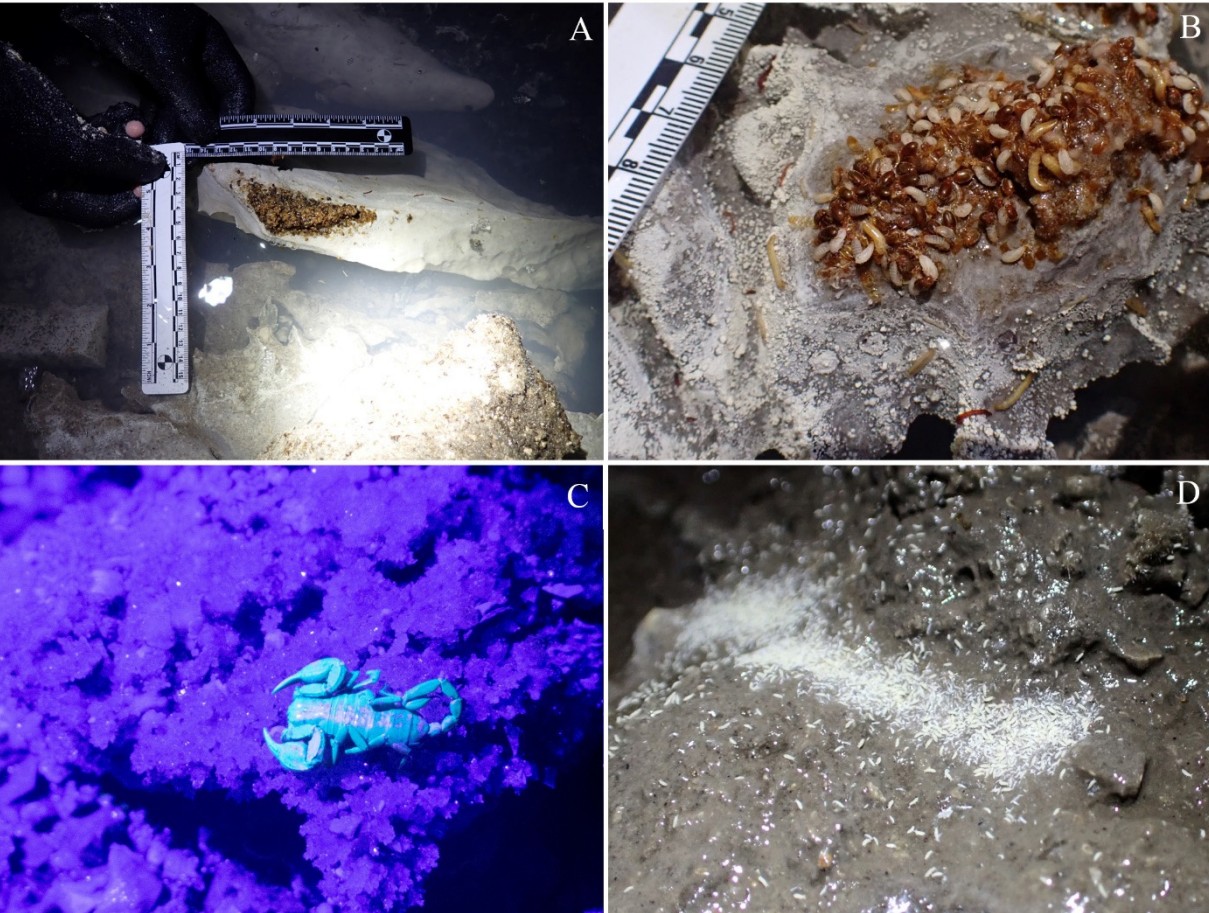

**Figure 3.** Particularities of methodology and cave fauna: (**A**) the method used to estimate the density of the larvae of *Chironomus* sp. (Chironomidae) and *Contacyphon palustris* (Scirtidae), i.e., counting individuals within a 15 × 15 cm quadrat; (**B**) dense agglomerations of yellow larvae, white apodous pupae, and brown adults of the *C. palustris* beetle observed on the rocky shores of the sulfidic stream; (**C**) adult specimen of *Euscorpius sulfur* (Euscorpiidae) found in the deep recesses of the cave, visualized using UV light; (**D**) dense agglomeration of collembola identified as an undescribed species of *Acheroxenylla* (Hypogastruridae) observed on thick brown biofilms covering the banks of small stagnant pools located in Sulfur Chapel.

## 3. Results

### 3.1. Microbial Communities

The white biofilms covering the bottom sediments in the cave stream were dominated by filamentous bacteria (e.g., *Thiotrix* and *Beggiatoa*) known as chemoautotrophic microorganisms that synthesize organic matter in situ using $H_2S$ as an electron donor and dioxygen from the cave atmosphere as an electron acceptor. Preliminary metagenomic investigations have shown that the brown biofilms covering the mud banks of the small pools and the blue biofilms on walls above the pools were dominated by chemosynthetic

sulfur-oxidizing microorganisms. The slightly blue biofilms on moist and anhydrite-free walls, where freshwater seeps in from the epikarst, were also predominantly composed of sulfur-oxidizing bacteria, particularly on the flowstone formations.

### 3.2. Vertebrate and Invertebrate Community

Two species of fish were observed in the sulfidic stream close to the entrance of the cave, i.e., an eel (*Anguilla* sp.) and numerous Cyprinid specimens (*Alburnoides* sp.). Seven species of bats belonging to three families (Rhinolophidae, Miniopteridae, and Vespertillionidae) (Table 1) were observed throughout the entire cave, including the most remote sections. The invertebrate fauna comprised five aquatic, 25 terrestrial, and four amphibiotic species (i.e., species having terrestrial adults and aquatic larvae). Of these, eight species were endemic and seven species exhibited troglomorphic traits (hereby considered as troglobites or stygobites) (Table 1).

**Table 1.** List of species found in Sulfur Cave, their habitat, identification method, and endemic status. Troglobites and stygobites are in bold.

| Higher Rank Taxon | Family | Species | Habitat | Identification | Endemic |
|---|---|---|---|---|---|
| Vertebrates | | | | | |
| Anguilliformes | Anguillidae | *Anguilla* sp. | aquatic | taxonomic | no |
| Cypriniformes | Cyprinidae | *Alburnoides* sp. | aquatic | taxonomic | no |
| Chiroptera | Rhinolophidae | *Rhinolophus ferrumequinum* (Screber, 1774) | terrestrial | taxonomic | no |
| Chiroptera | Rhinolophidae | *Rhiolophus hipposideros* (André, 1797) | terrestrial | taxonomic | no |
| Chiroptera | Rhinolophidae | *Rhinolophus blasii* (Peters, 1866) | terrestrial | taxonomic | no |
| Chiroptera | Rhinolophidae | *Rhinolophus euryale* (Blasius, 1853) | terrestrial | taxonomic | no |
| Chiroptera | Miniopteridae | *Miniopterus schreibersii* (Natterer in Kuhl, 1817) | terrestrial | taxonomic | no |
| Chiroptera | Vespertillionidae | *Myotis emarginatus* (É. Geoffroy Saint-Hilaire, 1806) | terrestrial | taxonomic | no |
| Chiroptera | Vespertillionidae | *Myotis bechsteinii* (Kuhl, 1817) | terrestrial | taxonomic | no |
| Invertebrates | | | | | |
| Oligochaeta | Naididae | *Tubifex tubifex* (O. F. Müller, 1774) | aquatic | taxonomic | no |
| Gastropoda | Lymnaeidae | *Radix labiata* (Rossmässler, 1835) | aquatic | molecular | no |
| Gastropoda | Hydrobiidae | *Grossuana euxina* (Wagner, 1928) | aquatic | molecular | no |
| **Amphipoda** | **Niphargidae** | ***Niphargus lourensis* Fišer, Trontelj & Sket, 2006** | **aquatic** | **molecular** | **no** |
| **Isopoda** | **Trichoniscidae** | **gen. sp.** | **terrestrial** | **taxonomic** | **yes** |
| **Isopoda** | **Trichoniscidae** | ***Alpioniscus* sp.** | **terrestrial** | **taxonomic** | **no** |
| Pseudoscorpiones | Neobisiidae | *Neobisium (Ommatoblothrus)* sp. | terrestrial | taxonomic, molecular | yes |
| **Pseudoscorpiones** | **Chthoniidae** | ***Chthonius* sp.** | **terrestrial** | **taxonomic, molecular** | **yes** |
| Scorpiones | Euscorpiidae | *Euscorpius sulfur* Kovařík et al., 2023 | terrestrial | taxonomic | yes |

**Table 1.** *Cont.*

| Higher Rank Taxon | Family | Species | Habitat | Identification | Endemic |
|---|---|---|---|---|---|
| Acarina | Astigmata | gen. sp. | terrestrial | taxonomic | no |
| Acarina | Labidostommatidae | *Eunicolina nova* Sellnick, 1931 | terrestrial | taxonomic | no |
| **Araneae** | **Leptonetidae** | ***Cataleptoneta* sp.** | **terrestrial** | **molecular** | **yes** |
| Araneae | Tetragnathidae | *Metellina merianae* (Scopoli, 1763) | terrestrial | taxonomic | no |
| Araneae | Agelenidae | *Tegenaria domestica* (Clerck, 1757) | terrestrial | molecular | no |
| Araneae | Nesticidae | *Kryptonesticus eremita* (Simon, 1880) | terrestrial | molecular | no |
| Araneae | Linyphiidae | *Prinerigone vagans* (Andouin, 1826) | terrestrial | taxonomic | no |
| Araneae | Linyphiidae | *Lepthyphantes magnesiae* Brignoli, 1979 | terrestrial | taxonomic | no |
| Chilopoda | Lithobiidae | *Lithobius viriatus* Sseliwanoff, 1880 | terrestrial | taxonomic, molecular | no |
| Chilopoda | Cryptopidae | *Scolopocryptops* sp. | terrestrial | taxonomic, molecular | yes |
| Chilopoda | Cryptopidae | *Cryptops hortensis* (Donovan, 1810) | terrestrial | taxonomic | no |
| **Collembola** | **Hypogastruridae** | ***Acheroxenylla* sp.** | **terrestrial** | **taxonomic** | **yes** |
| Collembola | Hypogastruridae | *Ceratophysella denticulata* (Bagnall, 1941) | terrestrial | taxonomic | no |
| Collembola | Onychiuridae | *Deuteraphorura* cf. *frasassii* (Fanciulli, 1999) | terrestrial | taxonomic | yes |
| Collembola | Entomobryidae | *Pseudosinella sexoculata* Schött, 1902 | terrestrial | taxonomic | no |
| Collembola | Entomobryidae | *Heteromurus nitidus* (Tempelton, 1836) | terrestrial | taxonomic | no |
| Collembola | Paronellidae | *Troglopedetes* sp. | terrestrial | taxonomic | no |
| Collembola | Sminthuridae | *Disparrhopalites patrizii* (Cassagnau & Delamare, 1953) | terrestrial | taxonomic | no |
| Collembola | Neelidae | *Neelus* sp. | terrestrial | taxonomic | yes |
| Diptera | Chironomidae | *Tanytarsus triangularis* Goetghebuer, 1928 | amphibiotic | taxonomic | no |
| Diptera | Chironomidae | *Chironomus* sp. | amphibiotic | taxonomic | no |
| Coleoptera | Scirtidae | *Contacyphon palustris* (C. G. Thomson, 1855) | amphibiotic | molecular | no |
| Coleoptera | Dytiscidae | *Hydroglyphus geminus* (Fabricius, 1792) | aquatic | taxonomic | no |
| Coleoptera | Hydrophilidae | *Coelostoma hispanicum* (Küster, 1848) | amphibiotic | taxonomic | no |
| **Coleoptera** | **Staphylinidae** | ***Tychobythinus* sp.** | **terrestrial** | **taxonomic** | **yes** |

*Chironomus* sp. (mean $\pm$ SD: 978 $\pm$ 1009; min–max: 45–2667) and larvae of *Contacyphon palustris* (1751 $\pm$ 2738; 45–9112) exhibited high densities in certain sections of the cave. The larval population of the scirtid beetle occurred on the rocky sediments in the swiftly flowing sulfidic stream near the springs. They share this habitat with numerous larvae of

*Chironomus* sp. (Chironomidae), the less frequent larvae of *Coelostoma hispanicum* (Küster, 1848) (Coleoptera, Hydrophilidae), and the adult aquatic beetle *Hydroglyphus geminus* (Fabricius 1792) (Coleoptera, Dytiscidae). When the *Chironomus* sp. larvae complete their development, they emerge from the aquatic environment and turn into adult flies. At this stage, they cease to feed, search for mates, reproduce, and die, often serving as food for various terrestrial predators inhabiting Sulfur Cave. Dense agglomerations of yellow larvae, white apodous pupae, and brown adults of the *C. palustris* beetle were observed on the rocky shores of the sulfidic stream (Figure 3B) and are likely preyed upon by numerous juvenile and adult specimens of *Euscorpius sulfur* Kovařík et al., 2023 (Euscorpiidae), which are often found in the deep recesses on the cave (Figure 3C).

Based on our observations, we approximated that more than 50,000 specimens of *Tegenaria domestica* (Clerck, 1757) (Agelenidae) inhabit the colony found on the cave wall near the cave entrance, along with several thousand specimens of *Prinerigone vagans* (Andouin, 1826), Linyphiidae. At the edges of this extended spider web, numerous individual spider webs host juvenile and adult specimens of *Metellina merianae* (Scopoli, 1763), Tetragnathidae, while an abundant population of *Lithobius viriatus* Sseliwanoff, 1880, Lithobiidae is found on the cave floor. Upon entering this cave section, a very dense population of chironomid flies, *Virgatanytarsus triangularis* (Goetghebuer, 1928), fills the air, and may represent a significant food resource for spiders and centipedes. Additionally, a very dense population of *T. triangularis* larvae is distributed across the rocky submerged sediments and may feed on the filamentous biofilms. The recently described endemic scorpion *Euscorpius sulfur*, occasionally occurs in Sulfur Cave and Turtle Cave [17].

The sulfidic cave stream in the deep sections of the cave is populated by eight aquatic species, three of which would leave the water upon completing their larval development. *Tubifex tubifex* (O. F. Müller, 1774) (Naididae) has a patchy distribution and was observed in areas with sandy sediments. The aquatic chironomid and beetle larvae may feed on the rich sulfur-oxidizing biofilms covering the stream sedimens, while *H. geminus* is likely to prey upon some of the young aquatic larvae. Two snail species were observed in the shallow flowing streams, scraping the bacterial biofilms: *Grossuana euxina* (Wagner, 1928) and *Radix labiata* (Rossmässler, 1835), both exhibiting sulfur deposits on the shells. The sole invertebrate species observed on the anhydrite deposits was the surface web-building spider *M. merianae* that likely fed on the chironomid flies. Instead, numerous Collembola belonging to the genus *Neelus* and small trichoniscid isopods were found on the moist surfaces of the flowstone formations created by the freshwater seeps in the cave. These organisms likely graze on microbial biofilms. In turn, they might be preyed upon by the eight predator species including the mite species *Eunicolina nova* Sellnick, 1931 (Labidostommatidae); two pseudoscorpion species, *Neobisium* sp. (Neobisiidae) and *Chthonius* sp. (Chthoniidae); two cryptopid centipede species, *Scolopocryptops* sp. and *Cryptops hortensis*, (Donovan, 1810); a pselaphin beetle, *Tychobytinus* sp.; and two web-building spider species, *Kryptonesticus eremita* (Simon, 1880) (Nesticidae) and the new troglobite species of *Cataleptoneta* sp. (Leptonetidae). Accumulations of guano on the cave floor attract terrestrial trichoniscid isopods belonging to the genus *Alpioniscus*, which may be preyed upon by *M. merianae*. A dense agglomeration of Collembola (Figure 3D) identified as an undescribed species of *Acheroxenylla* (Hypogastruridae) was observed on the thick brown biofilms covering the banks of small stagnant pools located in Sulfur Chapel.

*3.3. Carbon and Nitrogen Stable Isotope Analysis*

Preliminary stable isotope analysis showed light carbon and nitrogen stable isotope values for the terrestrial and aquatic invertebrates collected in Sulfur Cave. The organic carbon $\delta^{13}$C in these samples ranges between values of −27 and −32‰ compared with the standard PDB for carbon, and the organic nitrogen $\delta^{15}$N ranges between values of −3 and −10‰ compared with the standard of atmospheric air for nitrogen (Table 2).

**Table 2.** Nitrogen and carbon stable isotope values of selected taxa in Sulfur Cave.

| Sample ID | Taxon | Life Stage | $\delta^{15}$N Org ‰ | $\delta^{13}$C Org. ‰ |
|-----------|-------|------------|----------------------|------------------------|
| 01-2023 | *Lithobius viriatus* | adult | −3.37 | −31.47 |
| 02-2023 | *Lithobius viriatus* | adult | −3.97 | −32.00 |
| 03-2023 | *Lithobius viriatus* | adult | −4.07 | −32.08 |
| 04-2023 | *Contacyphon palustris* | larvae | −9.58 | −27.24 |
| 05-2023 | *Contacyphon palustris* | pupae | −9.72 | −27.66 |
| 06-2023 | *Contacyphon palustris* | adults | −9.75 | −28.12 |

## 4. Discussions

Similarly to other sulfidic subterranean ecosystems [6], Sulfur Cave is host to numerous different species that form stable subterranean populations. Thirty invertebrate species have been identified to date in this cave, of which eight are endemic: an amphipod, a scorpion, two pseudoscorpions, a spider, a springtail, a centipede, and a beetle, making Sulfur Cave a hotspot for subterranean diversity [18]. The presence of eight endemic species in Sulfur Cave (Table 1) suggests in situ evolution of the fauna due to secluded subsurface conditions. However, the number of endemic species is smaller than in Movile Cave [4] and Ayyalon Cave [19,20], and similar to Frasassi [21] and Cueva de Villa Luz caves [22,23]. The latter caves, having a closer connection with the surface fauna, thus sustain a smaller number of endemic species.

Surprisingly, however, most of the invertebrate populations encountered in this cave belong to surface species that do not display any troglomorphic traits such as reduction or loss of eyes and pigment, elongation of appendages [24]. This may be a consequence of the relatively small size of the caves in this region and their open access to the surface (they do not exceed a few hundred meters of subterranean passages), as well as the relative young age of the caves (hypogenic caves take a rather short time to develop; [25]), but we hypothesize the surface fauna is drawn to these caves by the presence of, and easy access to, significant amounts of food resources available year-round in the subterranean environment. The most surprising biological feature encountered in Sulfur Cave is an immense colonial spider web [10] spun by a very large population of the agelenid spider *T. domestica*. The species was previously documented in the entrance of dry caves [26] but has never been reported to build colonial nets.

The surprising abundance of the subterranean fauna encountered throughout Sulfur Cave and the presence of sulfidic water and ubiquitous microbial biofilms consisting of sulfur-oxidizing bacteria [27] suggests that this chemolithoautotrophic ecosystem is self-sustaining (i.e., the subterranean community relies on food produced in situ by chemoautotrophic microorganisms) [6,28]. Detailed genetic analysis of the cave microbiome will be the subject of future investigations conducted in Sulfur Cave and other sulfidic caves in this region. The results of the preliminary stable isotope analysis strongly support this hypothesis showing that negligible amounts of trophic resources of photosynthetic origin contribute to the base of the underground food web [20,29].

Sulfur Cave offers a unique opportunity to study the transfer of trophic resources from the aquatic to the terrestrial cave environment. We hypothesize that organic molecules generated chemoautotrophically by sulfur-oxidizing microorganisms forming biofilms that cover the bottom sediments in the cave stream are consumed by the aquatic larvae of two chironomids (*V. triangularis* and *Chironomus* sp.) and two beetle species (*C. palustris* and *H. geminus*). Upon completing their larval development, they emerge from the water and become aerial and terrestrial insects, respectively [30]. They cease to feed and most likely become the food base for the unusually abundant and diverse terrestrial cave community that consists almost exclusively of predators: 14 species of pseudoscorpions, scorpions, mites, spiders, centipedes, and beetles. Aquatic-to-terrestrial transfer of trophic resources is not uncommon in surface ecosystems [31], but the limited number of species involved in

this process in Sulfur Cave makes it easy to use this case study as a model in understanding this food transfer in more complex food webs that involve a multitude of species.

Biological studies currently performed in Sulfur Cave and other sulfidic caves in the Vromoner and Pixaria canyons on the Sarandaporo River and in the Langarica Canyon on a different tributary of the Aaos/Viosa River focus on subterranean biodiversity, ecology, and geomicrobiology. Ongoing research involves the continuation of the survey of the fauna inhabiting these caves and study of the spatial distribution of the different species in the subterranean environment. The results of the preliminary stable isotope studies led to the initiation of an ambitious isotopic survey of all the invertebrates reported from these caves, which is expected to shed light on their position within the subterranean food web. Morphological and molecular descriptions of new species are currently underway. Metagenomic methods are employed to analyze the species composition of the cave biofilms, determine their role within the cave microbiome, and attempt to decipher their contribution to limestone dissolution in the process of sulfuric acid speleogenesis.

## 5. Conclusions

The karst region at the border of Albania and Greece is unique due to its caves, which host aquatic and terrestrial-cave-restricted species associated with sulfidic subterranean aquifers and several cases of endemism. Given that this area is already impacted by human-driven changes, there is a pressing need for effective conservation strategies. Despite the establishment of national parks, this karstic area remains outside protected zones. Recognized by conservation initiatives such as "Save the Balkan Rivers" and "The Blue Heart of Europe", our results highlight the value of its subterranean biodiversity, which enhances its ecological significance and biological uniqueness, reinforcing its priority for conservation actions.

**Author Contributions:** Conceptualization, S.M.S.; validation, S.M.S. and T.B.; investigation, S.M.S., A.Ş. and R.I.B.; writing—original draft preparation, S.M.S., R.I.B., A.Ş. and T.B.; writing—review and editing, S.M.S., R.I.B., A.Ş. and T.B. All authors have read and agreed to the published version of the manuscript.

**Funding:** This research was funded by Biodiversa+, the European Biodiversity Partnership under the 2021-2022 BiodivProtect joint call for research proposals, co-funded by the European Commission (GA N°101052342) and with the funding organisations Ministry of Universities and Research (Italy), Agencia Estatal de Investigación—Fundación Biodiversidad (Spain), Fundo Regional para a Ciência e Tecnologia (Portugal), Suomen Akatemia—Ministry of the Environment (Finland), Belgian Science Policy Office (Belgium), Agence Nationale de la Recherche (France), Deutsche Forschungsgemeinschaft e.V. (Germany), Schweizerischer Nationalfonds (Grant N° 31BD30_209583, Switzerland), Fonds zur Förderung der Wissenschaftlichen Forschung (Austria), Ministry of Higher Education, Science and Innovation (Slovenia), and the Executive Agency for Higher Education, Research, Development and Innovation Funding (Romania) and partially supported by a grant of the Ministry of Research, Innovation and Digitization, CNCS—UEFISCDI, project number PN-III-P1-1.1-TE-2021-1047, within PNCDI III.

**Institutional Review Board Statement:** Not applicable.

**Data Availability Statement:** The original contributions presented in the study are included in the article. Further inquiries can be directed to the corresponding authors.

**Acknowledgments:** The authors would like to thank R. Nitescu, M. Hristescu, J-F. Flot, S. Flot, M. Kenesz, M. Fotiadi, M. Vaxevanopoulos, S. Galdenzi, A. Frumkin, N. Rosene, M. Audy, R. Bouda, C. Chauveau, M. Tawfeeq, I. Urák, A. Zsigmond, and A.D. Crînguş for their help during the field expeditions and V. Atudorei from the University of New Mexico in Albuquerque for his help with the preliminary stable isotope analysis. The identification of the specimens observed and collected in Sulfur Cave was performed by I. Urák (spiders), G. Popovici (centipedes and pseudoscorpions), M. Bertrand and P. Luptacik (mites), L. Kováč and V. Papáč (springtails), B. Klausnitzer and M. Toledo (beetles), E. Stur and T. Ekrem (chironomids), M. Patrick (oligochaetes), O. Popa (snails), F. Kovařík (scorpions), F. Stoch (amphipods), and G. Gentile (isopods). Special thanks to the Hellenic

Federation of Speleology, the Hellenic Institute of Speleological Research, and the Directorate General for the Protection and Development of Forests and the Rural Environment, Directorate of Forest Management of the Greek Ministry of Environment and Energy, for issuing research and sampling permits in caves in the Sarandaporo Valley, in compliance with the Nagoya Protocol on Access to Genetic Resources and the Fair and Equitable Sharing of Benefits Arising from their Utilization to the Convention on Biological Diversity.

**Conflicts of Interest:** The authors declare no conflicts of interest.

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
