# Peer review of "Exploring Biodiversity and Food Webs in Sulfur Cave in the Vromoner Canyon on the Greek–Albanian Border"

_diversity, doi:10.3390/d16080477_

Round 1

Reviewer 1 Report

Comments and Suggestions for Authors

On the example of Sulfur Cave the paper enriches our knowledge about regional cave biodiversity particularly based on chaemoautotrophic sources. It involves a wide range of methods, well written and illustrated. Further taxonomic investigation will refine the obtained results. Some minor remarks are included in the manuscript file.

Author Response

Dear Reviewer. We have addressed all changes and comments in the manuscript file reviewed by you. Thank you very much for that.

Reviewer 2 Report

Comments and Suggestions for Authors

This ms presents the results of biological surveys conducted from 2023 to 2024 in Sulfur Cave, in the Vromoner Canyon on the Greek-Albanian border, focusing on microbial, vertebrate, and invertebrate communities and investigating the structure of the subterranean food web. Areas of significant biological relevance were identified and reference data provided to inform conservation management of the biodiversity of this sulfidic cave.

The ms is generally well done, the conclusions supported by the data, along with appropriate tables and illustrations. Although the authors are not first language English speakers, the English is acceptable.

One thing that should be mentioned is where is all the material that was collected, deposited? It is important to provide the repositories for voucher materials for repeatability and verification purposes. Presumably, synoptic collections were also deposited in the two countries of origin as presumably stipulated in the permits?

Otherwise, there are only a few minor editorial corrections:

A font size error “Rhinolophidae, Miniopteridae and Vespertillionidae” on line 191 should be corrected.

Table 2 should appear on one page, not spread onto the second page (lines 264 and 265).

Author Response

Dear Reviewer. Thank you for seeing our manuscript. We have considered all changes and comments provided by you.
